# Narrative Review of the Safety of Using Pigs for Xenotransplantation: Characteristics and Diagnostic Methods of Vertical Transmissible Viruses

**DOI:** 10.3390/biomedicines12061181

**Published:** 2024-05-26

**Authors:** Su-Jin Kim, Joonho Moon

**Affiliations:** 1Apures Co., Ltd., 44, Hansan-gil, Cheongbuk-eup, Pyeongtaek-si 17792, Gyeonggi-do, Republic of Korea; sujinkim@apures.com; 2Division of Cardiology, Department of Internal Medicine, Seoul National University Hospital, 101, Daehak-ro, Jongno-gu, Seoul 03080, Republic of Korea

**Keywords:** assisted reproductive technique, designated pathogen-free facility, pig, vertical transmissible viruses, xenotransplantation

## Abstract

Amid the deepening imbalance in the supply and demand of allogeneic organs, xenotransplantation can be a practical alternative because it makes an unlimited supply of organs possible. However, to perform xenotransplantation on patients, the source animals to be used must be free from infectious agents. This requires the breeding of animals using assisted reproductive techniques, such as somatic cell nuclear transfer, embryo transfer, and cesarean section, without colostrum derived in designated pathogen-free (DPF) facilities. Most infectious agents can be removed from animals produced via these methods, but several viruses known to pass through the placenta are not easy to remove, even with these methods. Therefore, in this narrative review, we examine the characteristics of several viruses that are important to consider in xenotransplantation due to their ability to cross the placenta, and investigate how these viruses can be detected. This review is intended to help maintain DPF facilities by preventing animals infected with the virus from entering DPF facilities and to help select pigs suitable for xenotransplantation.

## 1. Introduction

There is a worldwide shortage of organs for transplantation. Xenotransplantation offers the prospect of an unlimited supply of organs and cells for clinical transplantation, thus resolving the critical shortage of human organs that currently prohibits a majority of patients on the waiting list from receiving transplants [1]. As a source animal for xenotransplantation, pigs are genetically, anatomically, and physiologically similar to humans, and have a lower risk of zoonosis than non-human primates. The survival rates in primate xenotransplantation experiments of porcine-derived cells, tissues, and organs are continuously increasing. In case of the heart transplantation, survival up to 945 days of heterotopic transplantation [2,3] and up to 264 days of orthotopic transplantation has been recorded [4]. Recently, an experiment was conducted in the United States to transplant the heart of a transgenic pig into a patient, and the report that the transplanted heart functioned normally for up to 60 days, surprised the world. Unfortunately, the patient died after being infected with a virus called porcine cytomegalovirus (PCMV), but it is still unknown whether this virus was the direct cause of death [5]. In the case of kidney xenotransplantation, there is a report of normal function for 758 days in monkeys [6], and recently, results of maintaining normal function for 3 days in the legs of a brain-dead patient have also been reported [7]. The longest survival is up to 14 days for lung xenotransplantation [8], whereas it is up to 29 days for liver xenotransplantation [9]. Compared to those of other organs, the maintenance periods of the lung and liver are short, but it is also an area where continuous efforts are made. In xenotransplantation of pancreatic islets, the maintenance of function for up to 603 days has been reported, despite the use of non-transgenic pig islets [10]. In the case of corneas, despite being non-transgenic porcine-derived corneas, it was reported that they functioned normally for up to 933 days [11]. It may be concluded that the transplantation of pig-derived cells, tissues, or organs has reached the level of feasibility.

On the other hand, the issue of zoonotic pathogens has emerged [12,13]. To obtain clinically applicable levels of xenogeneic cells, tissues, and organs from source animals, it must be assured that zoonotic infectious agents do not exist in the source animals. Currently, each country is defining or recommending designated pathogens. In Korea, 148 pathogens (48 species of bacteria, 13 species of parasites and helminths, 13 species of protozoa, and 74 species of virus) are recommended not to exist in source animals (i.e., pigs) as of 2022 by referring to the 2009 and 2016 IXA guidelines and the 2016 FDA guidelines. Among them is PCMV, a virus recently discovered in a patient receiving transgenic pig hearts. Therefore, animal-derived xenotransplantation products can be used only when it is confirmed that there is no PCMV in the source animal.

In our opinion, to breed source animals in a designated pathogen-free (DPF) facility, the most recommended method is to introduce them into DPF without feeding colostrum to offspring born through cesarean section (C-sec; which is a surgical procedure to deliver a baby through an incision made in the mother’s abdomen and uterus) at the specific pathogen-free (SPF) facility level. SPF facility refers to a facility where it has been proven that no specific pathogens generally affect animal health exist in experimental animals that live and grow there. On the other hand, DPF facilities should consider how pathogens affect recipients or human public health as well as how pathogens affect animal health because animals in DPF facilities breed for the source animal of xenotransplantation. However, if unavoidable, it is also possible to enter the DPF through C-sec after observing farm pigs under quarantine for a certain period (at least seven days). It is recommended to adopt a breeding program based on methods such as artificial insemination (i.e., a reproductive technique that involves the direct introduction of sperm into a female’s reproductive tract to achieve fertilization and conception), somatic cell nuclear transfer (i.e., a laboratory technique that involves transferring the nucleus of a somatic cell into an enucleated oocyte, used in animal cloning and stem cell research), embryo transfer (i.e., a technique that transfers the cloned embryos into a surrogate mother’s uterus to continue development and potentially produce offspring with identical genetic characteristics as the somatic cell donor), C-sec, and artificial milk to minimize the risk of transmission of infectious agents from mother to fetus. Nevertheless, for viruses known to pass through the placenta, it is necessary to determine the presence or absence of infectious agents through thorough examinations of the sows and to confirm that infectious agents do not exist by conducting the same test periodically in the offspring.

The placenta is a temporary organ that develops during pregnancy and helps the fetus in development. The placenta provides oxygen and nutrients to the fetus, removes carbon dioxide and waste products, produces hormones that help the fetus to grow, confers immunity, and protects the fetus from external stresses (e.g., physical shock or infection). The placenta of the eutherian mammal can be classified according to various criteria, such as the gross shape and histological structure of the maternal–fetal interface. If the placenta is classified by gross shape, it is divided into four types: diffuse (e.g., in horses and pigs), multicotyledonary (e.g., in ruminants), zonary (e.g., in carnivores), and discoidal (e.g., in primates, rodents, and rabbits). This is a morphological classification based on how the placenta is attached to the fetal membrane. Next, if the placenta is classified by histological structure, it is divided into three types: epitheliochorial (e.g., in horses, pigs, and ruminants), endotheliochorial (e.g., in carnivores), and hemochorial (e.g., in primates, rodents, and rabbits). As for the placenta of pigs, it belongs to the epitheliochorial type, consisting of three layers of maternal origin. On the other hand, the other two types of placenta are characterized by the partial or complete absence of layers on the maternal side [14]. In all three types of placenta, fetal structures always retain intact. Due to its multi-layered composition, the epitheliochorial type of placenta is least permeable to maternal substances, including pathogens [15]. Therefore, it is thought that the number and probability of infectious agents that can cross the placenta may be lower in pigs than in other animals [16]. This means that when transgenic pigs are produced through C-sec and put into DPF facilities, the number of infectious agents to be monitored may be relatively small compared to the case with other animals.

To date, the mechanism by which an infectious agent crosses the placenta to infect the fetus is unclear. In addition, since the discussion on this part is beyond the scope of this review, please refer to the paper on this issue [15]. In this review, we will discuss the characteristics of the viruses known to cross the porcine placenta and methods to detect them.

## 2. Methods

Xenotransplantation carries the risk of interspecies infection by various bacteria, viruses, protozoa, and parasites. To minimize this risk, the Ministry of Food and Drug Safety in Korea recently issued guidelines for xenotransplantation. The authors obtained clinical approval for xenogeneic pancreatic islet transplantation based on these guidelines, and in the process, we recognized the importance of understanding and detecting the nine viruses that pass through the placenta.

This narrative review primarily utilized PubMed and Google Scholar to search for data on these nine viruses using keywords such as the virus name, transplacental transmission, vertical transmission, (name of virus) detection methods, and (name of virus) vaccine. During the search process, no specific years were targeted. Information from sources such as the World Organization for Animal Health, Iowa State University, and the MSD Veterinary Manual were validated and utilized.

## 3. Viruses That Can Cross the Placenta

### 3.1. DNA Viruses

#### 3.1.1. Porcine Circoviruses (PCVs)

Porcine circoviruses (PCVs) are small icosahedral viruses with diameters of only 17 nm, containing a covalently closed circular, non-enveloped, and single-stranded DNA genome with a gene encoding a conserved replicase protein on the sense strand and one main capsid protein (Table 1) [17]. In pigs, four different PCVs have been identified and named with consecutive numbers based on the order of their discovery: porcine circovirus 1 (PCV1, first identified in 1974 [18]), porcine circovirus 2 (PCV2, first identified in 1997 [19,20,21,22,23]), porcine circovirus 3 (PCV3, first identified in 2016 [24]), and most recently porcine circovirus 4 (PCV4, first identified in 2019 [25]). PCVs are ubiquitous in global pig populations, and uninfected herds are rarely found. It is generally accepted that PCV1 is non-pathogenic [26,27]. PCV1 was isolated from stillborn piglets, and antibodies against PCV1 were identified in 10/160 fetal serum samples, indicating that vertical infections do occur occasionally [28]. PCV2 is considered an important economically challenging pathogen on a global scale as it causes postweaning multisystemic wasting syndrome (PMWS). Vertical transmission of PCV2 was confirmed by multiple laboratories [29,30]. Clinical disease associated with PCV2 infection can have multiple symptoms that include PMWS, porcine dermatitis and nephropathy syndrome (PDNS), porcine proliferative and necrotizing pneumonia (PNP), enteritis, reproductive failure, exudative dermatitis, and myocarditis/vasculitis [31,32]. Although PCV3 has attracted a lot of attention and research, its isolation from cell culture has only recently been successful [33], and its role as a pathogen remains controversial. PCV3 is widely spread as it has the ability to infect ticks [34] and mosquitoes [35], leading to various clinical symptoms, such as PDNS, porcine respiratory disease complex (PRDC), reproductive failure, enteric disease, central nervous system signs, including congenital tremors, and systemic inflammatory disease, mirroring the range of diseases associated with PCV2 (Table 1) [24,27,36,37,38,39,40,41,42,43,44,45]. In addition, the transmission of PCV3 from pig to baboon in pig-to-baboon heart xenotransplantation model was reported [46]. PCV4 was identified very recently in pigs with various health conditions from two farms in China [25]. Most of the characteristics of PCV4 are elusive, including its transplacental transmission ability. Therefore, PVC4 was excluded from this review.

Samples used for the detection of PCV1 are serum and superficial inguinal lymph nodes; for the detection of PCV2, they are serum, urine, feces, tracheobronchial swab, lymphoid organs (e.g., superficial inguinal lymph node), macrophage, monocyte, lung, endothelia, epithelia, and abortus; and for PCV3, they are serum, oral fluid, swab specimen from respiratory tract and biopsy, feces, semen, colostrum, heart, lung, and lymphoid tissue (Table 1).

The general PCV diagnosis of these antibodies has been demonstrated through indirect immunofluorescence or indirect immunoperoxidase tests. An antibody-detection enzyme-linked immunosorbent assay (ELISA) method has also been used for a large serological survey [47]. The production of monoclonal antibodies of PCVs can be used for immunohistochemistry (IHC) of cells and/or organs [28]. In addition, DNA-based methods, such as in situ hybridization (ISH), polymerase chain reaction (PCR), and quantitative PCR, were also developed; they were rapid and accurate. According to a report, PCV2 DNA can be detected in colostrum and milk in sow after delivery, as well as through serum and semen of boar via PCR [48]. In addition, the PCV2 antibody titer evaluation in colostrum and milk has been reported using anti-PCV2-IgG antibody levels, as determined by an open reading frame 2 (ORF2) capsid protein-based ELISA [49,50]. Furthermore, from reproductive organs and/or cells, the presence of PCV2 DNA and antigen has been demonstrated through IHC and ISH in sows and boars [30]. PCV3 also can be detected with PCR, quantitative PCR, and antibody test methods (e.g., ISH, IHC, or ELISA). PCV3 is a recently discovered virus, and it can also be detected through a fast and systematic visible method using the latest loop-mediated isothermal amplification (LAMP) method (Table 1) [51].

#### 3.1.2. Porcine Parvovirus (PPV)

Porcine parvovirus (PPV), a major causative agent of crucial reproductive failure in pigs, is a single-stranded DNA virus containing approximately 5000 nucleotides, non-enveloped, and having the ability of autonomous replication (Table 1). The genome of PPV has been identified through five major proteins, including three capsid proteins (VP1, VP2, VP3, and SAT) and two non-structural proteins (NS1 and NS2). VP1 and VP2 are responsible for the transcription and translation of the viral genome. VP3 is related to the modification of VP post-translation. SAT is a late non-structural protein, expressed from VP2 mRNA. Non-structural proteins, NS1 and NS2, are related to virus replication [52,53].

The first occurrence of PPV in pigs was described in 1967 [54]. It was isolated in association with herd infertility, abortions, and stillbirths in pigs [54]. Although in the initial infection (between 5 and 10 days), almost all infected pigs, independent of sex and age, showed transient leukopenia, the major clinical symptom in sow is reproductive failure. Precisely, the infected sows show symptoms such as embryo death before 35 days of pregnancy, mummification of fetuses of various sizes, increasing stillbirth, abortion, infertility, and unstable estrus cycle (Table 1) [16,55,56]. There is currently no reported evidence of zoonotic transmission of PPV. However, considering the rapid mutation rate and the ability of parvovirus to cross species barriers [57], it is thought to be necessary to consider the possibility of PPV infection in xenotransplant recipients undergoing prolonged immunosuppression.

After a sow is infected with PPV, the first event of viral replication occurs in its lymphoid tissues. Then, PPV systematically spreads out via viremia [58]. The transmission route of the virus from sow to fetus through the placental layers is not clear, as the six tissue layers perfectly divide the maternal to fetal blood circulation, which are accurately separated, and there is no chance the passing, even for small molecules and antibodies [16]. Furthermore, there is no evidence of viral replication in the uterine epithelium or in the trophoblast. However, it is known that when PPV infection occurs during the early stage of gestation, the virus can be transmitted to fetuses, leading to fetal death and mummification (Table 1) [59,60]. In a study by Miao et al. in 2009, PPV was detected using real-time PCR in several organs of PPV-challenged pregnant sows and their fetuses, with the highest viral load observed in the endometrium of the sows [61]. Although the exact mechanism of placental transmission remains unclear, there is a suggestion that maternal macrophages in the endometrium/placenta may play a role in the transplacental PPV infection. This is because many macrophages are present in the endometrium/placenta during gestation, and phagocytosis of PPV by macrophages has been observed. Although the possibility of transmission through the placenta by macrophages may be low, it can explain why transplacental PPV infection occurs only in a part of littermates [16,62]. The virus was suspected to use the endosomal pathways such as clathrin-mediated endocytosis or micropinocytosis for entrance in a cell [63]. Once inside the cell, the virus enters the nucleus and replicates its own DNA using the host’s replication machinery [64]. The replication of PPV damages the host’s cells and tissues by releasing toxic proteins and leading to apoptosis [65].

For several decades, PPV has been considered endemic to most pig herds worldwide. The infected animals release viruses for approximately two weeks through secretions and excretions. The virus can also spread through amniotic fluids and mummified fetuses from infected animals. More importantly, the virus is resistant to environmental influences, and contaminated facilities are the main sources of transmission.

For the diagnosis of PPV, several methods have been developed against different targets. In mummified fetuses and fetal remains, viral antigens were detected via immunofluorescence (IF). Virus antigen detection and/or virus titration were performed through a hemagglutination assay (HA) using several cell lines. Alternatively, serological diagnosis methods were used for PPV detection in blood or serum samples. Hemagglutination inhibition (HI) is one of the standard methods for quantifying virus antibodies. ELISA is another technique for the detection of PPV antibodies; it is a more convenient and sensitive method, and it is even faster and easier. More recently, DNA-based detection techniques have become more useful in detecting PPV. These techniques are more sensitive and fast. In addition, there is no limitation of samples for DNA-based detection as DNA can be extracted from fetal tissues, semen, and several other biological sources. Nowadays, DNA-based diagnosis techniques are advancing, and quantitative PCR and a LAMP assay are used for PPV detection (Table 1) [55,66,67].

#### 3.1.3. Porcine Cytomegalovirus (PCMV)

Porcine cytomegalovirus (PCMV) is a double-stranded DNA virus enveloped by an icosahedral protein. The virus was classified into the family Herpesviridae and the subfamily Betaherpesvirinae (Table 1) [68,69]. According to the present reports, the PCMV genome consists of three genes: DNA polymerase, major capsid protein (MCP), and glycoprotein B. Among these genes, MCP plays an essential role in the viral replication cycle. The MCP gene encodes vital structural proteins that make up the icosahedral capsid. In other reports, phylogenetic analysis based on the MCP gene and glycoprotein B gene sequencing has confirmed that the PCMV genome is more related to the genus Roseolovirus including human herpesvirus 6A, 6B, and 7 than to the Cytomegalovirus or Muromegalovirus genera, as other species of CMV (i.e., those from human, mouse, and rat) [69,70,71,72].

PCMV was first reported in 1955 by Done and lately has been found to be prevalent in all swine populations worldwide [73]. For piglets, it is a fatal pathogen, causing symptoms of inclusion body rhinitis (IBR), reproductive failure, pneumonia, anemia, and fever; however, in adult pigs, it generally leads to silent symptoms and/or subclinical infections (Table 1) [69,71].

Latent PCMV is usually harbored in monocytes/macrophages and CD8+ cells [74]. The first replication of PCMV breaks out in the nasal mucosa and/or lacrimal glands after infection. This is followed by cell-associated viremia two to three weeks post-infection and shedding of infectious virus in nasal secretions for a 10–30 day period [75,76]. Secondary replication differs by host age. In piglets, under weaning, the virus’s second replication occurs in the capillary endothelium and sinusoids of lymphatic tissues. For growing pigs, the virus has a tropism for nasal mucosal glands, lacrimal glands, kidney tubules, and, rarely, the epididymis and mucous glands of the esophagus [76,77]. PCMV can be transmitted through direct contact with nasal and ocular secretion, urine, and cervical fluid (Table 1) [78]. Vertical transmission of PCMV was observed in experimental infections, but there have been no confirmed cases in natural conditions [76,79,80]. In addition, a study by Egerer et al. in 2018 showed that PCMV-free herds can be established through early weaning [81]. Although it is possible to establish PCMV-free herds, more vigilant monitoring would be necessary because of the latent character of PCMV.

Clinical symptoms are commonly rare in all ages of swine, except from neonatal to young piglet and sow of childbearing age. Although PCMV-infected piglets exhibit shivering, sneezing, respiratory distress, and poor weighting, most piglets generally have not shown any clinical signs. After 3 weeks, PCMV is commonly subclinical to mild but is associated with other diseases and influences the infection of other viruses [82].

Clinical symptoms are rare in pregnant sows, and the infected sow exhibits symptoms for the first time only during late pregnancy. However, PCMV infection affects fetuses and may cause fetal death, mummified fetuses, stillbirths, weak babies, slight fever, inappetence, and nasal hemorrhage (Table 1) [83].

Recently, various methods for PCMV detection have been developed that target DNA, viruses, antigens, and antibodies. The virus could be isolated from epithelia of different tissues, such as testicular cells [84] or fallopian tubes [85]. The antigen was identified from tissue sections using histological methods such as staining [86]. As for the detection method using DNA, quantitative PCR, multiplex PCR, and LAMP have been developed since the detection of PCMV using the PCR method in 1999. The PCMV detection method using the ELISA method was developed in 1982 and has been widely used as a high-efficiency detection method [87]. Recently developed indirect-blocking ELISA method for the glycoprotein B epitope of the coating antigen has been shown to be highly specific and sensitive [88]. Western blot analysis method was also reported recently and it has been shown to have higher specificity compared to the ELISA method (Table 1) [89].

The samples used for detection are usually nasal swabs or whole blood in surviving individuals and turbinate mucosa, lungs, pulmonary macrophages (obtained via lung lavage), and kidneys in deceased individuals. In stillborn fetuses, the brain, liver, or bone marrow can also be used for viral detection (Table 1).

#### 3.1.4. Porcine Lymphotropic Herpesviruses (PLHVs)

Porcine lymphotropic herpesviruses (PLHVs) are double-stranded DNA viruses, classified into Suid gammaherpesvirus 3 (PLHV-1), Suid gammaherpesvirus 4 (PLHV-2), and Suid gammaherpesvirus 5 (PLHV-3) in the Herpesviridae family, Gammaherpesvirinae subfamily, and Macavirus genus because of their phylogenetic clustering with other members of this genus (Table 1) [90,91]. PLHV-1 and 2 were first reported in 1999 in pigs [92], and lately, PLHV-3 was first identified in 2003 [93]. The viruses were widespread in wild and/or domestic swine and the rate of their occurrence is comparable between healthy and diseased swine [94].

The Gammaherpesvirinae subfamily preferentially hosts lymphocytes or lymphoid tissue, having transformable ability, so this subfamily leads to lymphoproliferative disorders [95]. PLHVs were frequently detected in the blood and lymphoid organs, especially in B lymphocytes in swine [93]. Since their discovery, knowledge of their epidemiology has remained extremely limited. Transmission is thought to occur mainly horizontally; even the detection of PLHVs in cesarean-derived piglets suggests that vertical transmission, although rare, is possible [96]. PLHVs, especially PLHV-1, were related to post-transplant lymphoproliferative disease (PTLD) in the miniature pig experimental model [97]. Furthermore, PLHV-1 gene expression in PTLD pigs infected with PLHV-1 might lead to the etiology of lymphoproliferative disease [98]. However, there is no clinical report of natural infection of PLHVs from pigs to humans, but the knowledge of the clinical diseases is not clear. PTLD Porcine infected with PLHV-1 has shown various symptoms, such as lethargy, anorexia, high white blood cell count, and palpable lymph nodes, similar to human PTLD (Table 1) [99].

PLHVs have been detected in peripheral blood mononuclear cells (PBMCs), tonsils, liver, kidney, aorta, salivary gland, lung, and spleen [96,100,101]. The PCR assay was developed for the detection of PLHV, which targets a highly conserved region of the DNA polymerase (DPOL) gene in blood, and organ samples [92,93]. However, PCR is an insufficient method for PLHV detection, sometimes due to low sensitivity, even though some animals are infected, and identification of positivity infection in living animals is difficult [102]. Presently, quantitative PCR-based methods are possible for more sensitive detection of viruses than accomplished with conventional PCR. An antibody-based technique, ELISA is another detection method that analyzes the seroprevalence of PLHVs in infected animals; Western blotting also targets antibody response in pigs using the recombinant glycoprotein B (Table 1) [99,102].

**Table 1 biomedicines-12-01181-t001:** Vertical transmissible DNA viruses.

Name	Family	Characteristics	Infectious Species	Transmission Routes	Clinical Signs (Pig)	Sample Origin	Vaccines	Detection Methods	Clinical Signs (Human)	Ref.
PCV1	Circo-viridae	Single-stranded, non-enveloped, circular	Pig, cow, dog, rodent	Unknown	Non-symptomatic	Serum, superficial inguinal lymph node	Not commercially available	DNA-based tests (e.g., PCR), antibody tests (e.g., ISH)	-	[17,26,103,104]
PCV2	Circo-viridae	Single-stranded, non-enveloped, circular	Pig, cow, dog, rodent	Respiratory, digestive, and urinary secretion, direct contact, fomites (clothes, boots, equipment, etc.), blood	Postweaning multisystemic wasting syndrome (PMWS) porcine dermatitis and nephropathy syndrome (PDNS), porcine proliferative and necrotizing pneumonia (PNP), reproductive failure (e.g., abortion, reduced litter size, stillbirth), diarrhea, lymph node swelling, hemorrhage in skin and kidney, inflammation in skin and kidney, myocarditis/ vasculitis	Serum, urine, feces, tracheobronchial swab specimen, lymphoid organ (e.g., superficial inguinal lymph nodes), macrophage, monocytes, lung, endothelia, epithelia, abortus	Commercially available	DNA-based tests (e.g., PCR, quantitative PCR), antibody tests (e.g., ISH, ELISA)	-	[7,17,26,30,47,48,103,104,105,106,107,108]
PCV3	Circo-viridae	Single-stranded, non-enveloped, circular	Pig, cow, dog, deer, mouflon, tick, mosquito	Respiratory, digestive, and urinary secretion, direct contact, fomites (clothes, boots, equipment, etc.), blood, feces, tear, milk, semen	porcine dermatitis and nephropathy syndrome (PDNS), porcine respiratory disease complex (PRDC), reproductive failure, enteric disease, central nervous system sign (e.g., congenital tremor), multi-systemic inflammation	Serum, oral fluid, swab specimen from respiratory tract and biopsy, feces, semen, colostrum, heart, lung, lymphoid tissue	Not commercially available	DNA-based tests (e.g., PCR, quantitative PCR, LAMP), antibody tests (e.g., ISH, IHC, ELISA)	-	[7,17,27,34,35,51,109,110,111]
PPV	Parvo-viridae	Single-stranded, non-enveloped	Pig	Fomites (clothes, boots, equipment, etc.)	Reproductive failure (e.g., stillbirth, mummification, embryonic death, infertility (SMEDI), abortion, small litters, weak piglet)	Feces, urine, tissue, nasal swab, semen, intestine, lymphoid tissue	Commercially available	DNA-based tests (e.g., PCR, quantitative PCR, LAMP), serological tests (e.g., HA, HI) antibody tests (e.g., IF, ELISA)	-	[52,53,55,66,112]
PCMV	Herpes-viridae	Double-stranded, enveloped	Pig only	Nasal secretion, ocular secretion, urine, cervical fluid, direct contact, congenital transmission	Inclusion body rhinitis (IBR), reproductive failure (e.g., abortion, stillbirth, or mummification), pneumonia, anemia, fever	Semen, nasal swab, buffy coat, monocyte, lymphocyte, epithelia, lung lavage, kidney, Brain, liver, bone marrow	Not commercially available	DNA-based tests (e.g., quantitative PCR, LAMP), antibody tests (e.g., ISH, IHC, ELISA, WB)	-	[7,68,69,71,84,85,86,87,88]
PLHV	Herpes-viridae	Double-stranded, enveloped	Mostly in pig	Not clear, mainly horizontal and vertical transmission	Non-symptomatic post-transplant lymphoproliferative disease (PTLD), lethargy, anorexia, high white blood cell count, lymph node swelling	tissue, B lymphocyte, PBMC, tonsil, liver, kidney, aorta, salivary gland, lung, spleen	Not commercially available	DNA-based tests (e.g., PCR, quantitative PCR), antibody tests (e.g., WB, ELISA)	-	[90,91,92,93,94,96,98,99,100,101,102]

### 3.2. RNA Viruses

#### 3.2.1. Encephalomyocarditis Virus (EMCV)

The encephalomyocarditis virus (EMCV) was first isolated in 1945 by Helwig and Schmidt in Miami from a captive male gibbon that died suddenly of pulmonary edema and myocarditis [113] and subsequently from swine in Panama in 1958 [114].

The EMCV, similar to other picornaviruses, is a small non-enveloped virus, with an icosahedral capsid of 30 nm diameter. Its genome consists of a positive single-stranded RNA of approximately 7.8 kb that allows direct translation of the RNA into a polyprotein [115]. EMCV has been detected in many wild and domestic animals in many different areas around the world [116]. EMCV has a worldwide distribution and can infect a wide range of animal species. Its natural reservoir is thought to be rodents [117]. The presence of infected rodents in proximity to farms with infected swine also suggests that they play an important role in virus spread [118,119]. Although EMCV transmission from rodents to pigs is considered important, the impact of horizontal and vertical pig-to-pig transmission also needs to be quantified to understand their contribution to the spread of the disease on a pig farm (Table 2) [120].

EMCV has often been described as a potential zoonotic agent. Nevertheless, the association between human infection and disease has not yet been clearly established. Some experimental infections on human explants or human primary cell cultures have been described and clearly suggest that human cells are sensitive to EMCV [121,122]. In human cases in Peru, patients are presented with febrile illness (nausea, headache, and dyspnea), likely due to EMCV infection [123].

EMCV usually induces acute focal myocarditis with sudden death in pigs. Myocarditis is characterized by cardiac inflammation and cardiomyocyte necrosis. Other symptoms have been observed, such as anorexia, apathy, palsy, paralysis, or dyspnea. Susceptible pigs develop severe myocarditis followed by sudden death, while more resistant pigs develop mild myocarditis and can remain asymptomatic [124]. Histological analysis of piglet hearts reveals myocarditis associated with scattered or localized infiltration and accumulation of mononuclear cells, vascular congestion, edema, and myocardial fiber degeneration, with necrosis. In the brain, congestion is accompanied by meningitis, perivascular infiltration of mononuclear cells, and neuronal degeneration [125,126]. Reproductive disorders, including abortion, fetal death, and mummification, have been described in infected females (Table 2) [127].

Diagnosis assays for EMCV developed virus isolation, a microtiter serum neutralization test, IHC, ELISA, and a different series of nucleic acid testing assays (NATs). All the above methods have played important roles in the diagnosis of EMCV infection. However, virus isolation, microtiter serum neutralization tests, and IHC are laborious and time-consuming, and are not usually used in the laboratory for routine diagnosis. At present, NAT assays, including the conventional reverse-transcription PCR (RT-PCR), quantitative RT-PCR based on SYBR Green or TaqMan probe, and reverse-transcription LAMP (RT-LAMP), are widely accepted for the routine detection of EMCV in the laboratory. In recent years, recombinase polymerase amplification for detecting EMCV in swine was reported (Table 2) [128].

#### 3.2.2. Hepatitis E Virus (HEV)

The hepatitis E virus (HEV) is a small, spherical, non-enveloped, and positive-sense single-stranded RNA virus with icosahedral capsid symmetry. It belongs to the genus orthohepevirus in the family hepeviridae (Table 2) [129]. It was named later than the other hepatitis viruses, but its existence was actually known in the early 1980s [130]. The family hepeviridae is divided into two genera: orthohepevirus, which contains HEVs that affect mammalian and avian species, and pescihepevirus, which contains HEVs affecting cutthroat trout. The orthohepevirus genus is further divided into four subgenera, depending upon the species affected. Orthohepevirus A contains the HEVs infecting humans, pigs, wild boars, deer, mongoose, rabbit, and camel. Orthohepevirus B contains chicken HEV. Orthohepevirus C contains the HEVs that infect rats, shrews, ferrets, and mink. Orthohepevirus D contains HEV isolated from bats [131]. Within orthohepevirus A, there are at least seven different genotypes. HEV-1 and HEV-2 are host-restricted to humans, while HEV-3 and HEV-4 are zoonotic, with swine serving as the reservoir. HEV-5 and HEV-6 have been found in wild boar, and HEV-7 has been found in dromedary camels [129]. As a zoonotic agent, swine HEV is an emerging public health concern in many industrialized countries. Generally, hepatitis E is a self-limiting disease, but it can become chronic or cause severe disease in immunocompromised patients or those with a history of liver or chronic diseases [132].

Pigs are a natural reservoir for HEV, and the consumption of raw or undercooked pork is an important source of foodborne HEV transmission. Occupational risks such as direct contact with infected pigs also increase the risk of HEV transmission in humans. Cross-species infections of HEV-3 and HEV-4 have been documented under experimental and natural conditions. Both swine HEV-3 and swine HEV-4 infect non-human primates and humans. Swine HEV, predominantly HEV-3, can establish chronic infection in immunocompromised patients, especially in solid organ transplant recipients. The zoonotic HEV-3 and, to a lesser extent, HEV-4 have also been shown to cause neurological diseases and kidney injury [133].

The pathogenesis of HEV in swine is largely unknown, and information is limited. After fecal–oral infection of young swine (2–3 months old), shortly after maternal antibody wanes, viremia occurs, lasting 1–2 weeks [134]. Fecal viral shedding occurs 1–2 weeks after inoculation but may persist for up to eight weeks [135]. Infection has been reported in one-month-old pigs [136]. Pigs are susceptible to infection, but natural and experimental infections in swine are asymptomatic; they do not suffer clinical disease [137,138], In reproduction, HEV in swine has shown more effects on litter size or preterm abortions in pregnant gilts; however, more examination is required (Table 2) [135].

In humans, HEV usually manifests as acute hepatitis. Acute hepatitis E caused by HEV1 or HEV2 has a latent period of about 40 days, showing various clinical symptoms, from asymptomatic to fulminant hepatitis, with a mortality rate of about 1%. Typical symptoms include fatigue, anorexia, nausea, vomiting, fever, abdominal pain, and jaundice (Table 2). Aminotransferase and bilirubin are elevated and usually normalize within six weeks and are accompanied by some neurological complications [139]. According to reports, neurological complications occur in 8% of hepatitis E patients. The most common diseases are Guillain–Barre syndrome, neuralgic amyotrophy, and meningitis/encephalitis [140]. HEV-3 and HEV-4 infections can progress to chronic hepatitis in immunosuppressed patients. Chronic hepatitis E has been reported in patients with solid organ transplants, HIV-infected patients with low CD4+ T cells, and patients with hematologic cancers receiving chemotherapy [141,142]. If chronic hepatitis develops, rapid liver fibrosis may occur, leading to cirrhosis, decompensated liver failure, and death in some patients [139].

Recognizing HEV infection in swine is difficult because of the absence of clinical signs. Any pig can be infected with HEV, and natural infection seems to most commonly occur at the time maternal immunity wanes in piglets around 7–9 weeks of age [133]. HEV nucleic acids have been traditionally detected using semi-nested RT-PCR [143]. Primers amplify a segment of the HEV ORF2 gene, which encodes the immunogenic capsid protein [136]. The most widely used assay for the detection of nucleic acids in human HEV infections is a quantitative RT-PCR TaqMan assay, which uses primers from ORF3 and can detect genotypes HEV-1 to HEV-4. This assay can detect swine HEV, even in environmental samples, which is more sensitive than nested RT-PCR [144]. HEV isolation is difficult; the virus is uncultivatable in cell culture. However, recombinant capsid proteins can be expressed using *Escherichia coli* containing a cloned capsid gene and purified from hyperimmunized animals for diagnostics [145]. IHC has been used to detect HEV-3 antigens in tissues and characterize the type and severity of the inflammatory process by binding to CD3 surface receptors on porcine immune cells. Viral antigens have been detected using rabbit anti-HEV-3 hyperimmune serum with recombinant capsid proteins [146]. ISH has also been used to detect HEV RNA in tissues [136]. A commercially available ELISA kit has been developed. PrioCHECK^®^ HEV Antibody ELISA kit uses HEV-1 and HEV-3 antigens encoded by ORF2 and ORF3 to detect antibodies. The sensitivity and specificity of the commercially available ELISA kit are both greater than 90% [147]. Other ELISA tests have been developed with specificity and sensitivity greater than 95% using capsid protein antigens, anti-HEV serum, and rabbit anti-porcine IgG [145]. A Western blot method has been reported for the detection of antibodies reacting to the HEV capsid protein (Table 2) [148].

#### 3.2.3. Porcine Reproductive and Respiratory Syndrome Virus (PRRSV)

PRRSV is an enveloped single-stranded positive-sense RNA virus belonging to the family Arteriviridae in the order Nidovirales (Table 2) [149]. All PRRS virus (PRRSV) isolates are classified into two genotypes: PRRSV-1 and PRRSV-2 [150]. The PRRSV genome is approximately 15 kb in length and contains at least 10 ORFs, a short 5′ untranslated region, and a poly(A) tail at the 3′ terminus [151]. ORF1a and ORF1ab encode the replication-related polymerase proteins and are processed into at least 13 non-structural proteins through self-cleavage, and the other ORFs encode eight structural proteins [152]. PRRSV exists as two major genotypes, the European prototype (EU-type, type 1), known as the Lelystad virus (LV), and the North American prototype (NA-type, type 2), known as VR-2332 [149]. Strains LV and VR-2332 share about 55–70% nucleotide identity at the genome level and about 50–80% amino acid similarity [153]. PRRSV can be transmitted by insects, including house flies [154] and mosquitoes [155].

Porcine reproductive and respiratory syndrome (PRRS) has become an economically critical factor in the global swine industry since it was first reported in the United States in 1987 [156,157]. Breeding age gilts, sows, and boars show clinical signs that may include a period of anorexia, fever, lethargy, depression, and perhaps respiratory distress or vomiting. Mild cyanosis of the ears, abdomen, and vulva has been reported in some outbreaks. Reproductive problems, often the most obvious signs, include a decrease in the number of dams that conceive or farrow. There is usually an increase in premature farrowing, late-term abortions, stillborn or weak piglets, and mummified fetuses. Preweaning mortality is high. Nursing pigs may have dyspnea. The period for reproductive signs varies with herd size but is usually two to three months in duration. In larger operations, signs may be cyclical, especially if naïve gilts or sows continue to be introduced into the herd. There is evidence that subpopulations within large breeding herds escape initial infection but are infected when exposed later and serve as sources of continued virus shedding. In addition, herds may be infected with multiple heterologous strains of the PRRSV that are not completely cross-protective. In boars, clinical signs are similar to sows and are accompanied by a decrease in semen quality.

The primary clinical signs among young pigs are fever, depression, lethargy, stunting due to systemic disease, and pneumonia. Sneezing, fever, and lethargy are followed by expiratory dyspnea and stunting. The peak age for respiratory disease is 4–10 weeks. Postweaning mortality is often markedly increased, especially with more virulent strains and the occurrence of ever-present concurrent and secondary infections. Older pigs have similar respiratory signs. Heterologous infections may lead to prolonged or repeated outbreaks of respiratory disease (Table 2).

Serum or semen is one of the convenience samples for the detection of PRRSV in animals and/or herds [158]. Usually, DNA-based detection methods, such as RT-PCR and quantitative RT-PCR, can be used for the detection of viruses from these specimens. These DNA-based detection methods are faster and more accurate than others. According to another recent study, oral fluid can replace serum/semen for PRRSV diagnostics not only giving a similar efficiency and cost-effectivity but also losing the burden in terms of labor, time, and safety of both workers and animals [158]. Antigen or antibody tests can be also applied to detect PRRSV, such as IHC and ELISA. Specifically, many diagnostic methods of PRRSV have been developed using antigen–antibody immune reaction and immunochromatographic tests via two gold-labeled monoclonal antibodies [159]. Fluorescence resonance energy transfer assay, detecting the signal between the fluorescent dye and the quantum dot or gold nanoparticle [160] and on-site differential diagnostic detection strip, was developed using fluorescent probe-based immunoassay (Table 2) [161].

**Table 2 biomedicines-12-01181-t002:** Vertical transmissible RNA viruses.

Name	Family	Characteristics	Infectious Species	Transmission Routes	Clinical Signs (Pig)	Sample Origin	Vaccines	Detection Methods	Clinical Signs (Human)	Ref.
EMCV	Picornaviridae	Single-stranded, non-enveloped	Pig, Primate, Even-and Odd-toed ungulate, Elephant, Carnivore, Rodent	Rodent-to-pig transmission, direct contact, nasal secretion, feces, urine	Encephalitis, myocarditis, sudden death, anorexia, apathy, palsy, paralysis, dyspnea, mummification poor conception rate, embryo resorption, stillbirth, abortion, neonatal death	Heart, liver, kidney, spleen tissue from acutely dead animal or abortus	Commercially available	RNA-based tests (e.g., quantitative RT-PCR), antibody tests (e.g., ELISA)	-	[115,128,162]
HEV	Hepe-viridae	Single-stranded, positive sense, non-enveloped	Pig, human, deer, ruminant, rabbit, horse, donkey, mule	Oro-fecal transmission, direct contact, contaminated food, Blood transfusion (human to human), organ transplantation (human to human)	Non-symptomatic	Feces, bile, blood, liver, small intestine, lymph mode, colon	Commercially available (highly protective for humans, and not been tested in swine)	RNA-based tests (e.g., RT-PCR, quantitative RT-PCR), antibody tests (e.g., ISH, IHC, ELISA, WB, EIA for specific IgM)	Fever, headache, fatigue, nausea, vomiting, abdominal pain, diarrhea, jaundice	[7,129,136,143,144,145,146,147,148,163,164,165,166]
PRRSV	Arteri-viridae	Single-stranded, positive sense, enveloped	Pig	Semen, aerosol transmission, direct contact, fomites (clothes, boots, equipment, etc.), insects, house flies	Porcine reproductive and respiratory syndrome (PRRS), blue ear pig disease, reproductive failure (e.g., abortion, stillborn, mummification), anorexia, fever, lethargy, depression, vomiting, cyanosis, lymph node swelling, nephritis, myometritis, endometritis	Blood, saliva, semen, feces, nasal secretion, milk, tonsil, lung, lymph node, tissue sample of acutely fallen ill pig or weak-born piglet	Commercially available	RNA-based tests (e.g., RT-PCR, quantitative RT-PCR), sequencing, antibody tests (e.g., IHC, ELISA)	-	[7,149,158,160,161,167]

## 4. Strengths and Limitations

The strength of this review lies in its provision of explanations regarding the definitions, transmission routes, clinical symptoms, sampling sites, detection methods, and commercial vaccine availability for the nine viruses considered crucial in the context of xenotransplantation. However, this paper has the limitation of not providing predefined PCR primer sequences or antibodies. This limitation arises from the variability of viral strains regionally and temporally, necessitating customization for each laboratory based on the specific region or time. Furthermore, the optimal detection method for certain viruses remains a subject of debate among experts [89,168], leading to limitations in presenting definitive testing methods in this paper.

## 5. Conclusions

This is a literature that demonstrates that six DNA viruses and three RNA viruses could pass from surrogates to piglets through the placenta. The DNA viruses are PCV1, PCV2, and PCV3 belonging to circoviridae; PPV belonging to parvoviridae; and PCMV and PLHV belonging to herpesviridae. The RNA viruses are EMCV belonging to picornaviridae, HEV belonging to hepeviridae, and PRRSV belonging to arteriviridae. Among them, PPV, PCMV, PLHV, and PPRSV are known to cause infection mostly in pigs, and five viruses that could be cross-contaminated with other animals are PCV1, PCV2, PCV3, EMCV, and HEV. As mentioned above, PCV3 can specifically infect ticks and mosquitoes, and PRRSV can be transmitted by insects, including house flies and mosquitoes, suggesting the importance of insect management in breeding facilities. HEV, in particular, is a zoonotic virus that spreads to humans. Unlike pigs, which do not show any special clinical symptoms, humans show clinical symptoms (e.g., fever, headache, malaise, nausea, vomiting, abdominal pain, diarrhea, and jaundice) upon infection. This necessitates the prevention of transmission to humans when testing for infectious agents in pigs in breeding facilities.

PCV1 and HEV are known to elicit no specific symptoms in pigs, whereas seven other viruses have been confirmed to cause distinct clinical symptoms in pigs. Viral infection can be diagnosed through clinical symptoms, but symptoms may not arise in the source animals due to different levels of susceptibility and the viral load. A virus belonging to the herpesviridae family (e.g., PCMV and PLHV) also may not cause symptoms in the animals used for xenotransplantation. Hence it is important to use a diagnostic method sensitive enough to detect infections in such cases so that the possibility of transmission of a virus to humans upon transplantation can be minimized. Currently, it is recommended to test most of the nine viruses using DNA-based methods, and in the case of viruses capable of latent infection, it is also recommended to test antibodies against infectious agents, such as Western blot [89]. A general consensus needs to be reached on testing methods and the frequency of testing for the nine types of vertical transmissible viruses. To this end, a meeting of the Cellular, Tissue, and Gene Therapies Advisory Committee (CTGTAC) was held in June 2022, and active discussions between experts and regulatory authorities took place.

We need to be prepared for a future scenario in which a pathogen capable of vertical transmission is added to the list of the nine known vertically transmissible viruses. New pathogens are expected to emerge due to the climate and environmental changes. Therefore, the field of xenotransplantation will require a continual update through active discussions of the pathogens in source animals.

## Data Availability

Data are contained within the article.

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
