# Peer review of "Narrative Review of the Safety of Using Pigs for Xenotransplantation: Characteristics and Diagnostic Methods of Vertical Transmissible Viruses"

_biomedicines, 2024, doi:10.3390/biomedicines12061181_

Round 1

Reviewer 1 Report

Comments and Suggestions for Authors

This review is useful for clinical xenotransplantation.  The author described a practical way how to maintain clean pig in DPF by cesarean-section in SPF facility.  They also pointed out  DNA and RNA viruses which might be passed through the placenta systemically.

However, the references of 3 (Xenotransplantation 2022) and 5(AJT 2019) in Introduction might be changed to Längin M, et al. Nature 2018 and Anand RP, et al. Nature 2023. So, the explanation would be changed.

Author Response

To Reviewer 1,

Thank you for taking the time to read our paper thoroughly and providing thoughtful comments. Your review has helped us identify aspects we may have overlooked and areas where improvements are needed, which will undoubtedly enhance the quality of our paper. We are grateful for your valuable feedback.

We will break down the commented parts into separate questions, explaining or making revisions for each one, and then provide explanations for those specific sections.

This review is useful for clinical xenotransplantation. The author described a practical way how to maintain clean pig in DPF by cesarean-section in SPF facility. They also pointed out DNA and RNA viruses which might be passed through the placenta systemically.

→ We appreciate your positive evaluation of my paper. The requested revisions have elevated the paper to a higher standard.

However, the references of 3 (Xenotransplantation 2022) and 5(AJT 2019) in Introduction might be changed to Längin M, et al. Nature 2018 and Anand RP, et al. Nature 2023. So, the explanation would be changed.

→ If we understand correctly, we have determined that the reference "Längin M, et al. Nature 2018" might be placed after reference 2, and we have added it accordingly. Regarding reference 5, we were unable to verify the newly released paper. we have accordingly revised the reference and content to align with the provided 'Anand RP, et al. Nature 2023'. Your confirmation would be appreciated.

If we have misunderstood or misinterpreted the reviewer's intentions, please let me know, and we will make the necessary corrections.

Thank you once again for taking the time to review my manuscript, and we appreciate your feedback.

Reviewer 2 Report

Comments and Suggestions for Authors

Manuscript ID: biomedicines-2973846

Type of manuscript: Review

Title: Safe pigs for xenotransplantation: Characteristics and diagnostic methods of vertical transmissible

The aim of this review is to discuss the characteristics of the viruses known to cross the porcine placenta and methods to detect them.

Comments and Suggestions for Authors:

The manuscript is an interesting review, but requires some considerations.

Title and Abstract: It should be indicated that this is a narrative review.

A section should be included with the methodology with which the review was carried out: eligibility criteria for the selection, Information sources and search strategy.

The authors do not comment on any the strengths and limitations of the review.

Author Response

To Reviewer 2,

Thank you for taking the time to read our paper thoroughly and providing thoughtful comments. Your review has helped us identify aspects we may have overlooked and areas where improvements are needed, which will undoubtedly enhance the quality of our paper. We are grateful for your valuable feedback.

We will break down the commented parts into separate questions, explaining or making revisions for each one, and then provide explanations for those specific sections.  

Type of manuscript: Review

Title: Safe pigs for xenotransplantation: Characteristics and diagnostic methods of vertical transmissible

The aim of this review is to discuss the characteristics of the viruses known to cross the porcine placenta and methods to detect them.

Comments and Suggestions for Authors:

The manuscript is an interesting review, but requires some considerations.

→ We appreciate your positive evaluation of my paper. The requested revisions have elevated the paper to a higher standard.

Title and Abstract: It should be indicated that this is a narrative review.

→ We have added "Narrative review of" at the beginning of the title.

A section should be included with the methodology with which the review was carried out: eligibility criteria for the selection, Information sources and search strategy.

The authors do not comment on any the strengths and limitations of the review.

→ The comment provided has made a significant contribution to enhancing the completeness of the manuscript by addressing the overlooked aspects. Accordingly, the following content has been added to lines 111-133 of the manuscript:

  1. Methods

Xenotransplantation carries the risk of interspecies infection by various bacteria, viruses, protozoa, and parasites. To minimize this risk, the Ministry of Food and Drug Safety in Korea recently issued guidelines for xenotransplantation. The authors obtained clinical approval for xenogeneic pancreatic islet transplantation based on these guidelines, and in the process, we recognized the importance of understanding and detecting the nine viruses that pass through the placenta.

This narrative review primarily utilized PubMed and Google Scholar to search for data on these nine viruses using keywords such as the virus name, transplacental transmission, vertical transmission, (name of virus) detection methods, and (name of virus) vaccine. During the search process, no specific years were targeted. Information from sources such as the World Organization for Animal Health, Iowa State University, and the MSD Veterinary Manual were validated and utilized.

  1. Strengths and Limitations

The strength of this review lies in its provision of explanations regarding the definitions, transmission routes, clinical symptoms, sampling sites, detection methods, and commercial vaccine availability for the nine viruses considered crucial in the context of xenotransplantation. However, this paper has the limitation of not providing predefined PCR primer sequences or antibodies. This limitation arises from the variability of viral strains regionally and temporally, necessitating customization for each laboratory based on the specific region or time. Furthermore, the optimal detection method for certain viruses remains a subject of debate among experts, leading to limitations in presenting definitive testing methods in this paper.

Please review the added paragraphs and let us know if it meets your expectations.

If we have misunderstood or misinterpreted the reviewer's intentions, please let me know, and we will make the necessary corrections.

Thank you once again for taking the time to review my manuscript, and we appreciate your feedback.

Reviewer 3 Report

Comments and Suggestions for Authors

The manuscript did not add any novelty in the field.

The title should be modified accorging to the aim and the content of  the text as reported in lines 15-16 "Therefore, in this review, we look at the characteristics of several viruses known to cross the placenta and determine how they can be detected. ": this means that the focus is the detection of the viruses for diagnostic purposes and the xenotransplantation is not considered at all. Otherwise, staring from the title the reader is quite confused. No data were reported about the xenotransplantation being this merely speculative and again confusing the reader.

Comments on the Quality of English Language

Typing/grammar errors and organization of the text in terms of repetition and coherence of the sentences should be improved.

Author Response

To Reviewer 3,

Thank you for taking the time to read our paper thoroughly and providing thoughtful comments. Your review has helped us identify aspects we may have overlooked and areas where improvements are needed, which will undoubtedly enhance the quality of our paper. We are grateful for your valuable feedback.

We will break down the commented parts into separate questions, explaining or making revisions for each one, and then provide explanations for those specific sections.

The manuscript did not add any novelty in the field.

The title should be modified accorging to the aim and the content of  the text as reported in lines 15-16 "Therefore, in this review, we look at the characteristics of several viruses known to cross the placenta and determine how they can be detected. ": this means that the focus is the detection of the viruses for diagnostic purposes and the xenotransplantation is not considered at all. Otherwise, staring from the title the reader is quite confused. No data were reported about the xenotransplantation being this merely speculative and again confusing the reader.

→ Thank you for providing us with the opportunity to address our oversight. Would it be helpful to reduce confusion among readers by revising the 15-16 sentences, instead of changing title, as follows: “Therefore, in this review, we examine the characteristics of several viruses that are important to consider in xenotransplantation due to their ability to cross the placenta, and investigate how these viruses can be detected.”?

Typing/grammar errors and organization of the text in terms of repetition and coherence of the sentences should be improved.

→ Correcting typos and grammar errors is not always easy for us as non-native speakers. May we proceed with making simple edits with native English-speaking acquaintances while going through the revision process and seek professional editing at the end? The paper has already undergone one round of professional English editing. We seek the reviewer's understanding.

If we have misunderstood or misinterpreted the reviewer's intentions, please let me know, and we will make the necessary corrections.

Thank you once again for taking the time to review my manuscript, and we appreciate your feedback.

Round 2

Reviewer 2 Report

Comments and Suggestions for Authors

Round 2.

 The authors make changes to the manuscript following indications that improve it.

 There would be some aspects to consider:

- Abstract: It should be indicated that this is a narrative review (line 19).

- The Strengths and Limitations section would be better placed before the Conclusion section.

- In the added limitations it is commented that "the optimal detection method for certain viruses remains a subject of debate among experts." A bibliographical citation could be provided to support this assertion.

Author Response

Reviewer Comments:

To Reviewer,

Thank you for taking the time to read our paper thoroughly and providing thoughtful comments. Your review has helped us identify aspects we may have overlooked and areas where improvements are needed, which will undoubtedly enhance the quality of our paper. We are grateful for your valuable feedback.

We will break down the commented parts into separate questions, explaining or making revisions for each one, and then provide explanations for those specific sections.

Round 2.

 The authors make changes to the manuscript following indications that improve it.

 There would be some aspects to consider:

- Abstract: It should be indicated that this is a narrative review (line 19).

→ I apologize for overlooking the points you mentioned in previous round of revision process. I have now included the term "narrative" where you indicated. Could you please confirm if it has been inserted correctly? Thank you for your understanding.

- The Strengths and Limitations section would be better placed before the Conclusion section.

→ I have moved the 'Strengths and Limitations' section to the position you indicated and have adjusted the numbering of each section accordingly. Could you kindly confirm if everything is in order? Thank you.

- In the added limitations it is commented that "the optimal detection method for certain viruses remains a subject of debate among experts." A bibliographical citation could be provided to support this assertion.

→ Dr. Joachim Denner made this assertion at the Cellular, Tissue, and Gene Therapies Advisory Committee (CTGTAC) meeting held in June 2022, which sparked an intense discussion among experts. The presentations from that meeting were subsequently published as papers, which I have cited in the references. The relevant references are listed below.

89. Halecker, S.; Hansen, S.; Krabben, L.; Ebner, F.; Kaufer, B.; Denner, J. How, where and when to screen for porcine cytomegalovirus (PCMV) in donor pigs for xenotransplantation. Sci Rep 2022, 12, 21545.

168. Denner, J. Virus Safety of Xenotransplantation. Viruses 2022, 14.

If we have misunderstood or misinterpreted the reviewer's intentions, please let me know, and we will make the necessary corrections.

Thank you once again for taking the time to review my manuscript, and we appreciate your feedback.
